# University Students’ Knowledge, Attitudes, and Behaviors Related to Marine Environment Pollution

**DOI:** 10.3390/ijerph192416671

**Published:** 2022-12-12

**Authors:** Yongtao Gan, Jian Gao, Jiahao Zhang, Xia Wu, Tian Zhang, Mengjun Shao

**Affiliations:** 1Institute of Higher Education, Shantou University, Shantou 515063, China; 2Jing Hengyi School of Education, Hangzhou Normal University, Hangzhou 311121, China

**Keywords:** marine environment pollution, knowledge, attitude, behavior

## Abstract

Recent research on marine environment pollution (MEP) has primarily focused on legislative and market-based instruments rather than on understanding related knowledge, attitudes, and behaviors. Within this context, we used a survey of university students in China to investigate attitudes and behaviors related to MEP. Specifically, we employed a tri-component attitude model to analyze questionnaire data from 446 randomly selected students. Our results indicate that participants had a good knowledge of MEP. Furthermore, our data revealed the following three MEP-related attitudinal clusters: activists, supporters, and onlookers. Activists showed negative attitudes toward MEP with strong anti–MEP behaviors. Supporters also had negative attitudes toward MEP but performed less anti–MEP behaviors. Finally, onlookers exhibited indifferent attitudes with neutral MEP-related behaviors. Each of the three attitudinal clusters varied according to the demographic characteristics of the participants. The implications of these results on the reduction in MEP were discussed.

## 1. Introduction

The ocean is an important part of the Earth’s ecosystem that houses millions of organisms. Against this background, the Chinese government places substantial significance on the development of its marine economy and the protection of its vast marine territory, long coastline, and rich marine resources [1]. The relationship between the health of the coastal environment and human livelihoods, described as ocean citizenship, requires detailed study to better understand coastal and marine-related issues [2]. Although the ocean is essential for human survival, it is most threatened by anthropogenic activities. As a result of runoff from rivers, impermeable surfaces, and storm drains from severe weather events, almost 80% of marine debris is transported from the land to the ocean [3,4]. For example, plastic products, one of the primary sources of marine environment pollution (MEP), cannot degrade and remain afloat on the sea for extended periods, leading to the deterioration of seawater quality and ingestion by seabed organisms.

As college students do not pay much attention to MEP, they do not always understand its causes [5]. These causes include the physical pollution of the natural environment, such as from natural waste products. Although the government has imposed bans and implemented policies for reducing the consumption of single-use plastics along with coastal clean-up efforts undertaken by civic authorities, for litter prevention programs to be effective, management barriers, littering behavior, social norms, context-specific litter dynamics and characteristics, and pollution sources need to be addressed [4]. Thus, awareness regarding behaviors and attitudes toward plastic pollution should be instilled, as consumption patterns and lifestyles in addition to other factors contribute to MEP [6].

Marine ecosystems and their related services have been heavily impacted by changes in ocean chemistry and increasing average sea surface temperatures. With numerous industry–related pollutants ending up in the ocean, the ability of the ocean to self-regulate has been severely affected. For example, chemical elements, with varying degrees of harmfulness, pose a great threat to the stability of the entire Earth’s ecosystem.

Although MEP is currently relevant, many people are not aware of its importance and possible related solutions. Against the background of ocean citizenship and the ever-growing concerns surrounding the marine environment, several studies have investigated perceptions of coastal and marine issues [7]. Furthermore, certain international instruments and iconic books including the Earth Summit (2012), World Summit on Sustainable Development (2002), Rio Declaration of Environment and Development (1992), Our Common Future (1987), The Stockholm Conference (1972), and Silent Spring (1962) have proven to be milestones in shaping and influencing attitudes toward the environment. From the beginning of the 21st century, countries worldwide have begun to pay attention to the dire consequences of MEP. In view of human dependency on the ocean, the need to support and protect the marine environment has become increasingly urgent.

### 1.1. MEP-Related Knowledge

Various studies have focused on the perceptions and attitudes of university students related to MEP [5,8]. Normative knowledge of MEP underlies university students’ attitudes toward it and results in corresponding environmental behavior, thereby ensuring the sustainability of the marine environment [9]. Within this context, researchers reported that university students become aware of sustainable behaviors after understanding the characteristics of resources and related benefits [10]. Notably, Kotowicz et al. [11] showed that MEP-related knowledge plays a crucial role in the development of socio-ecological knowledge. However, having MEP-related knowledge does not automatically translate into conservation-oriented behaviors, nor does it always correspond to attitudes aligned with the knowledge [12]. On the contrary, several studies have found attitude to be inconsistent with related behavior [13].

In a study by Boubonari et al. [9], Greek pre-school teachers were reported to have a moderate knowledge of MEP and a positive attitude toward the marine environment. In addition, previous study suggested that Chinese students’ information about the marine environment was primarily obtained from mainstream media [14]. However, mainstream media frequently relies on buzzwords, often reflecting a lack of education and interpretation, and can contain mistakes. In contrast, formal education in schools accounts for only a small proportion of students’ knowledge. Hence, to promote pro-environmental behavior, a shift in values, a better understanding of the role of behaviors related to marine environment, and an enhanced awareness of environmental marine issues are essential [15].

In Chinese contemporary school education, transmitting relevant marine knowledge to students has become an important measure to protect the marine environment. Within this context, Ref. [16] investigated the curriculum, behavior, attitudes, and knowledge related to marine environment protection of senior grade Taiwanese primary school students. Notably, examples and studies on the knowledge, attitude and behavior of MEP are limited.

### 1.2. Attitudes toward MEP

Shaped through worldviews, situations, or objects, attitude can be described as a latent socio-psychological feeling contained and nurtured within the self [10]. Nonetheless, despite the recent interest in studies on MEP, few studies have focused on perceptions and attitudes related to it.

Although numerous studies have investigated public attitude toward MEP, these studies are limited in scope and depth, focusing solely on respondents’ concerns about its impact [9]. Negative or positive perceptions related to marine pollution, or the environment predict corresponding attitudes [6,17]. While some studies have demonstrated the development of public perceptions around MEP (from negative to positive) in certain industrial societies in the U.S., Germany, and France, such developments have yet to be recorded in Asia and Africa [6,18,19,20]. Owing to the overall lack of marine education in many developing countries, the general perception of MEP can be described as largely indifferent or pro-active [17]. These indifferent or pro-active attitudes are accepted because the concept of MEP forms part of the daily lives of most citizens in developing countries.

Notably, whether or not an individual acts upon his/her environmental concerns regarding MEP issues depends on the individual’s perceptions of their ability and capacity to make an effective difference in mitigating these problems [21]. Against the background of the cognitive judgment of citizens regarding environmental protection, research findings indicate that in Greece, people are expected to display positive attitudes toward MEP [22].

At the attitude level, we intuitively understand the different attitudes of current college students toward MEP through the survey data. On one hand, it evaluates the results of marine education, on the other, it provides a reference for the formulation and promulgation of marine education policies.

### 1.3. Behaviors Related to MEP

Behavior can be described as the expression and overt action toward a referent [23]. Furthermore, beliefs, along with the covert and innate feeling of social psychology toward a referent, are often reflections of behavior [16,24]. In exploring the influence of attitude on behavior, social psychology studies indicate the inextricable link between them [12]. This finding highlights the need to understand the attitudes underlying specific behaviors, as changes in behavior can be brought about by changes in attitude [25].

Attitude is the primary driving force of behaviors related to the ocean [26]. However, changing behaviors is a multi-dimensional and complex issue with several factors. Specifically, the behavior of an individual toward MEP and the attitude underlying that behavior can be affected by non-personal factors such as economic, political, social, and environmental contexts, together with personal factors within the control of the individual [27]. Although behavior is influenced by several non-personal factors such as habits, social norms, and contextual support, the attitude remains the major influencing factor [28]. Notably, behavior and environmental attitudes can be influenced by affective (values and attitudes), cognitive (environmental awareness), and demographic (education, age, and gender) factors. Regarding demographic factors, women are considered more sensitive to the environment than men. However, this phenomenon is not consistent throughout the literature, with some studies reporting men to be more environmentally sensitive than women [27,29].

At the behavioral level, not only can we intuitively understand college students’ environmental behaviors toward MEP, but we can also explore the differences in cognition and behavior through follow-up analysis. Because marine education requires implementation and differences between attitudes and behaviors often arise, exploration of behaviors can provide a better direction for the development of the marine economy.

### 1.4. Research Objectives

Our study builds on the findings of previous and related studies by analyzing the results of a survey of university students based on self-reported knowledge, attitudes, and behaviors related to MEP in China. We selected university students as they are considered to be decision-makers and educated members of future society [30]. Furthermore, university students are more likely to become future community leaders and public opinion influencers [8]. Within this context, the knowledge, attitudes, and behaviors of these students toward MEP, may have a significant impact on MEP decision-making in China.

Understanding the amount, source, and scientific nature of college students’ knowledge of MEP is essential in identifying the factors affecting college students’ acquisition of MEP knowledge to provide direction and reference for future college education on MEP. The main objective of this paper is to explore college students’ knowledge, attitude, and behavior toward MEP as they are about to enter the social construction group. In order to better cope with the current situation of MEP, formulate corresponding solutions, and develop a more efficient marine economy, researchers need to understand the group’s cognition and behavior toward it. This information would be valuable in the encouragement of behavioral changes and interventions to reduce harmful actions toward the marine environment.

## 2. Material and Methods

### 2.1. Theoretical Framework

We used a three-component model of attitude based on Grimm [31] as the theoretical basis of our study. Composed of the three interrelated components of cognition, affection and behavior, the model assumes attitude to be a socio-psychological structure. As attitude is the perception of a referent or object, its cognitive component refers to an individual’s belief or knowledge toward a referent. The behavior component of the model is based on the manner in which an individual’s attitude influences behavior [31]. Hence, in light of the cognitive and emotional components, behavior can be described as an observable response. Furthermore, based on the nature and type of the affective and cognitive components of attitude, the response, or behavior, similar to the other two components, can be either negative or positive.

In our study, the perceptions and/or beliefs held by the university students regarding MEP were assumed unique to the study participants, thereby constituting them as the basis for categorization. Despite beliefs and feelings having an influence on behavior, individual responses related to the different attitude components will result in different behaviors toward MEP. In addition to providing us with a basis for describing the study participants, the three-component model offers an analytical and theoretical tool to understand behaviors and attitudes related to MEP.

### 2.2. Data

Data were collected online from November 2021 to January 2022 at three universities in Guangdong province, China. Two of the three sample universities are located on the coast. We used a questionnaire survey with stratified random sampling to obtain data on university students’ knowledge, attitudes, and behaviors toward MEP. A total of 446 completed questionnaires were used.

Our research was approved by the relevant person in charge of the first author’s university, and prior informed consent was obtained from all participants. Participation in the research study was voluntary and participants remained anonymous. Several social media platforms were used to invite as many participants as possible. After prior informed consent was given, participants were guided through an online questionnaire survey. Overall, participants took approximately 5 to 10 min to complete the survey. To ensure the validity of the research results, participants needed to fulfill the following requirements: (1) participants had to be enrolled at a college or university at the time of completing the survey, and (2) each participant could only complete one questionnaire to avoid repetition and redundancy. Data of participants that had not been exposed to relevant content in marine education were excluded from further analyses.

Based on the three-component attitude model, we investigated students’ knowledge, attitudes, and behaviors related to MEP. In the design process of this study, behavior related to marine environment protection was set as the dependent variable. According to results from previous studies, awareness of the marine environment included related education, knowledge, attitudes, and behaviors [32]. Hence, based on the importance of knowledge and attitudes toward marine conservation, we investigated the relationship between marine conservation behaviors, knowledge, and related attitude. Within this context, we selected behavior as the dependent variable.

The questionnaire survey was structured and designed to obtain as much objective information from the participants as possible. Both open- and close-ended questions were used in the questionnaire survey, consisting of the following parts:Demographic profile of respondents;Knowledge component;(1)The concept of MEP;(2)Types of MEP;(3)Understanding the effects of MEP;(4)Understanding MEP incidents;(5)Expertise in MEP;(6)Understanding the causes of MEP;

Knowledge related to MEP was evaluated and used to develop a ten-point scale. We based the scale on findings of previous studies toward attitudes related to the environment [9,23,27,33,34], designed to provide an overall indication of participants’ attitudes to MEP. Participants were asked to rate statements on a Likert scale from 1 (Strongly Disagree) to 5 (Strongly Agree).

### 2.3. Data Analyses

Data entry, editing, cleaning, and analysis were conducted using SPSS version 24. In addition, we used a Chi-square test of independence, one-way ANOVA, and two-step clustering for further data analysis.

The two-step clustering technique was used to classify participants’ survey data based on attitudes toward MEP, using the log-likelihood distance measurement of the Schwarz Bayesian Criterion. Using this technique, the system tools divided the topics with similar load values on the MEP attitude scale into separate categories. We named these according to the overall characteristics of the topics in each category to guide the subsequent data analysis. According to the results of clustering and the comprehensive naming of similar feature topics, we obtained three cluster names of marine environmental protection attitudes: activists, supporters, and onlookers.

To understand the knowledge and behavior traits associated with each attitudinal component, one-way ANOVA was used to analyze the three attitude components (cognition, behavior, and emotion). Finally, to identify patterns and relationships between the attitudinal components based on the demographic characteristics of the participants, we used a Chi-square test of independence. This analysis was conducted to identify and understand the different clusters, with the aim of reducing MEP by supporting and encouraging specific behavioral, attitudinal, and educational interventions.

## 3. Results

### 3.1. Demographic Profiles

Both male (46.6%) and female (53.4%) students participated in the study (Table 1). Notably, a little over half of the participants (50.2%) were in their first year and 11.7% were in their second year of university. The majority of participants (81.2%) were located at a coastal university, and approximately one-third (38.3%) selected a coastal city as their hometown. Although approximately one-quarter (25.6%) of participants completed university-level marine-related courses, over two-thirds (64.3%) indicated participation in marine-related activities.

### 3.2. MEP-Related Knowledge

Results from our survey indicated that university students had relatively high levels of MEP-related knowledge (score of 7.22). However, “Expertise in MEP” only acquired a score of 4.81, with “Understanding the causes of marine pollution” obtaining a score of 4.72 (Figure 1). Furthermore, the value of Cronbach’s alpha showed an acceptable internal consistency of MEP-related knowledge (α = 0.83).

### 3.3. Attitudes toward MEP

We compared the average scores of participants that corresponded to each mean value in terms of cognition, emotion, and behavior toward MEP. Our results indicated that participants had more negative attitudes toward MEP than negative MEP-related behaviors (Table 2).

We investigated various clusters of attitudes (Table 3) and compared these clusters using a sample level element score. In addition, we examined the characteristics of every cluster based on the uniqueness of the response to each element of the cluster. Cluster 1 displayed a higher score than Cluster 2 and 3, indicating that participants within Cluster 1 were the most in favor of reducing MEP. Cluster 2 showed an intermediate and higher level of attitude with high scores recorded for two items within the cognitive dimension of attitude (item 1 = 4.01, item 3 = 4.23). As Cluster 3 indicated low scores on all the elements of attitude, participants within that this cluster could be described as not being concerned about MEP.

### 3.4. Attitudes toward MEP and Related Knowledge Traits

Consistent with their highly negative attitude toward MEP, participants of Cluster 1 received the highest scores for all MEP knowledge items. Cluster 2 showed the next highest scores, and Cluster 3 had the lowest. However, only two knowledge items had significant differences between the three clusters (Table 4), namely “Types of MEP” and “Understanding of MEP incidents”.

### 3.5. Behavioral Traits Associated with Attitudinal Segments

To better understand participants’ attitudes toward MEP, we compared relevant attitudes with related behavioral traits (Table 5). Together with being genuinely concerned about MEP, participants in Cluster 1 showed positive behaviors on all behavioral elements. Furthermore, in terms of behavioral items, participants within this cluster demonstrated a significant tendency toward behaviors related to active reduction in MEP. Therefore, as Cluster 1 consisted of participants who do not support MEP and who are even willing to spend to participate in MEP-reducing activities, we designated Cluster 1 as “Activists”.

In terms of reducing MEP, Cluster 2 participants showed somewhat less significant unfavorable behaviors, with a moderate concern regarding MEP, as evidenced by the scores obtained for all behavioral elements. Nevertheless, participants in this cluster supported MEP-reducing behavior, although not as strongly as the participants of Cluster 1. In addition, the MEP concern (cognitive aspect) of participants of Cluster 2 were not reflected in their emotions (affective component). Thus, our results indicate that participants of Cluster 2, although not completely concerned about MEP, could, depending upon the situation, practice related behaviors. As participants within Cluster 2 have the potential to become pro-conservation activists, we renamed this cluster the “Supporters”.

In addition to having the lowest MEP-related attitude scores, participants of Cluster 3 are described as onlookers of MEP. Therefore, we renamed this cluster the “Onlookers.” Cluster 3 displayed the lowest scores in all MEP-related behavior elements (Table 3).

### 3.6. Demographic Profile of Attitudinal Segments

We used a Chi-square test of independence based on the demographics of the participants to identify the profiles of the three MEP-related attitudinal segments (Table 6). Regarding the seven demographic characteristics of the participants, our results indicated that all three attitudinal segments had different statistical significance (Table 6). Of the activists, female participants constituted a higher proportion (57.10%) than male participants (47.10%) (Table 6). However, a higher proportion of supporters were males (45.20%) compared with females (54.80%). A significant number (68.00%) of participants that completed marine-related subjects were activists. In addition, a higher proportion of first year students were activists (61.6%) than students who have been at university for longer. Notably, more than half of the participants (56.1%) that completed courses related to the ocean are activists. Furthermore, while over half of participants based at coastal city universities can be described as marine environment activists (54.1%), only 48% of participants that grew up in coastal cities were described as activists.

## 4. Discussion

Our findings indicated that university students who participated in our survey had a good understanding of MEP. This knowledge can lead to the development of more environmentally aware attitudes. Participants’ attitudes toward MEP were divided into three separate segments, namely the activists, supporters, and onlookers. As students’ attitudes were found to be heterogeneous toward MEP, a reduction in MEP would require various changes in the related attitudes and behaviors.

Based on the assumption of homogeneity in the general public’s attitude toward MEP, several interventions and policies have been identified to reduce it [27,35,36]. However, using a “one size fits all” approach does not consider the variations in respondents’ behaviors and attitudes, as shown by the findings of this study. From an environmental psychology perspective, despite having similar social backgrounds, the respondents in our study had unique reactions toward similar environmental issues [17,37].

Our results indicate a gap between the knowledge and attitudes of respondents. Specifically, “activists” received the highest scores for all MEP-related items of knowledge, followed by “supporters,” and then “onlookers.” Although participants were knowledgeable about MEP, they had less expert knowledge about its causes, consequently affecting their MEP-related behaviors. Specifically, the students’ behaviors have significant impact on the quality of the marine environment. Knowledge on MEP is obtained through environmental education and higher education through tertiary institutions. While the awareness of behavior stems from knowledge, behaviors related to the conservation of the marine environment stem from the actions of acting, thinking, and feeling a sense of responsibly toward it [9,27].

In our study, the various cognitive, affective, and behavioral responses to MEP are represented by the different attitudinal and behavioral segments. On one end, the “activists” believe MEP to be a significant problem and express these feelings through related behaviors, with highly unfavorable cognitive and conative attitudes toward MEP. Moreover, when necessary, these students are willing to voice their concerns. In addition, this segment represents students that are not willing to put up with MEP and reflect pro-environmental behaviors. On the other, the “onlookers” are unable to express their concerns about MEP. Thus, the members of this segment can be described as the direct opposite of the “activists,” with more neutral cognitive and affective attitudes toward MEP. In addition, the “onlookers” are less concerned about the harmful consequences of pollution on both marine and terrestrial lives. The attitude and behavior of members of this category can be described as indifferent to MEP.

The “supporters” form the third segment are located between the two extremes of “activists” and “onlookers.” Not openly expressing their dislike of MEP, the “supporters” have high cognitive responses to MEP, although with more neutral attitudes. In addition, the pro-conservation behavior of the “supporters” is not as strong as that of the “activists.” Although the “supporter’s” behavior is more inclined toward the “activists” than the “onlookers,” they are placed between the two other segments because of their mixed characteristics. Despite having the potential to become “activists,” to transform their cognitive pro-conservation beliefs into conservation-related behaviors, “supporters” need ongoing MEP-related education with clear examples of its negative consequences [38,39].

In our study, all three MEP attitudinal segments varied according to the demographic characteristics of the participants. This finding shows that further attitude and behavior toward MEP are based on background characteristics. For example, Erhabor and Don [5] as well as Amoah and Addoah [40], found that students must understand the threats posed by MEP to behave as supporters and activists.

Our results, which revealed more female participants than male in the MEP activist segment, suggests that future behavior change strategies should incorporate sex differences. In general, because the activist segment does not support MEP, these participants have the potential to reduce MEP. Therefore, to create behavioral changes, increasing the proportion of activists in all grades and subjects is necessary. Specifically, our findings, which show that more females are pro-conservation activists than males, suggest that future behavioral change strategies should include a sex aspect.

In addition, our discovery that a high proportion of students who participated in ocean-related courses or activities are “activists” corresponds to findings that pro-environmental behavior is related to the level of education received [41,42]. Within this context, awareness of environmental issues among young people in China can be attributed to education and exposure to the marine environment [5,40]. In China, marine education can inspire the younger generation to cultivate pro-conservation attitudes and behaviors [40]. The results of our study indicate that the proportion of MEP activists increases with an increase in marine-related education. This corresponds to our results that a higher proportion of first year students are activists than students who have been at university for more than a year. In addition, a higher proportion of coastal university students are activists compared with students from inland universities. These findings suggest that coastal universities are more focused on marine education and MEP-related knowledge than inland universities [43].

## 5. Research Limitations

The study was only conducted in three universities in Guangdong, China. As it primarily focused on the attitude of coastal college students toward MEP, rather than with coastal students less exposed to the sea, because of the limited research sample collection, the research value is small. Future studies should consider an expanded sample that includes coastal college students and non-coastal college students to compare their MEP-related knowledge, attitudes, and behavior.

## 6. Conclusions

Although interest in reducing MEP is increasing, limited empirical research has focused on obtaining a better understanding of the behavior, attitude, and knowledge of university students toward it. Within this context, our study investigated and categorized university students in China based on their behaviors and attitudes, with the aim of identifying behavioral changes and interventions to reduce MEP. Several conclusions can be drawn from the results of our study.

University students were heterogeneous in their knowledge, attitude, and behavior toward MEP, contrary to the belief that pro-conservation behaviors are homogeneous and that a “one size fits all” solution could be used to reduce MEP. From our survey results, we identified three segments of behaviors and attitudes of the university students, categorized as “activists”, “supporters”, and “onlookers”. Every segment displayed a unique behavior and attitude toward MEP. The different reactions and sentiments of the public toward MEP need to be considered in the development and implementation of programs and policies targeted at reducing it, particularly those focused on behavioral changes. More effective and sustainable results can be achieved by such targeted efforts. For instance, to specifically influence attitudes of each segment, discussions with “onlookers” and “supporters” on specific behavioral changes should be adopted.

The results indicated that encouraging pro-environmental attitudes and behaviors necessitates further and continuous education that highlights the threats of MEP. Social mediums along with both formal and informal marine education, such as the addition of community programs targeted for adults and illiterate members of society and the inclusion of MEP issues in university curricula and formation of pro-environment clubs, could help cultivate and develop attitudes and behaviors that lead to the conservation of the marine environment.

Moreover, considering the variation of the three investigated attitudinal segments across demographic characteristics, we suggest that interventions for behavioral changes should be based on demographic findings. For example, first year students exhibited more pro-environment attitudes and behaviors compared to students who had been at university for longer. Hence, more effort is needed to influence the behavior of these students. In addition, to increase the proportion of activists and sustain pro-environmental behavior in younger generations, specific topics regarding MEP should be included in the educational curriculum.

## Figures and Tables

**Figure 1 ijerph-19-16671-f001:**
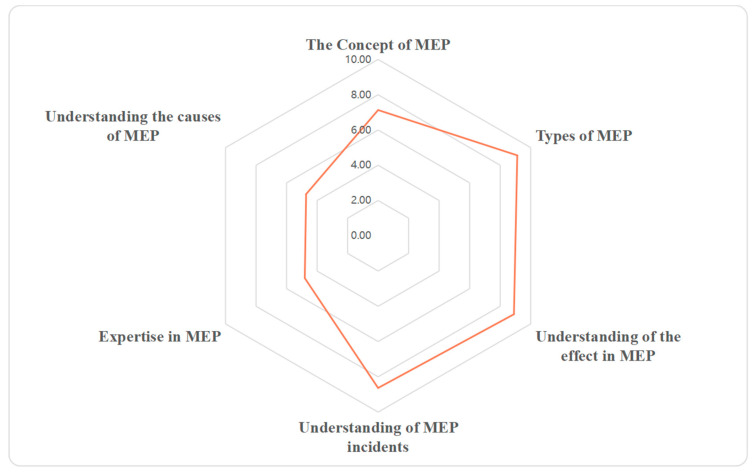
University students’ knowledge about marine environment pollution (MEP).

**Table 1 ijerph-19-16671-t001:** Demographic characteristics of participants.

Demographic	Frequency	Percentage (%)
Sex		
Male	208	46.6
Female	238	53.4
Subject		
Social Science	118	26.5
Engineering	189	42.4
Nature Science	65	14.6
Marine-related	25	5.6
Other	49	11
Year level at university		
First year	224	50.2
Second year	52	11.7
Third year	88	19.7
Fourth year	82	18.4
Is your hometown located on the coast?		
Yes	171	38.3
No	275	61.7
Is your university located on the coast?		
Yes	362	81.2
No	84	18.8
Have you taken marine-related courses in university?		
Yes	114	25.6
No	332	74.4
Have you participated in marine-related activities in university?		
Yes	287	64.3
No	159	35.7

**Table 2 ijerph-19-16671-t002:** University students’ attitudes toward marine environment pollution (MEP).

Attitude toward MEP	Mean	Standard Deviation	Corrected Item-Total Correlation	Cronbach Alpha
Cognitive				
I think single-use plastics are harmful to marine ecosystems.	4.30	0.76	0.81	0.91
I think marine environment pollution is currently a serious problem.	4.20	0.77	0.79	
I think marine environment pollution negatively affects our tourism fortunes.	4.48	0.73	0.84	
To reduce the amount of solid waste in the ocean, we should reduce plastic production.	4.13	0.90	0.68	
I think daily-life waste products cause marine pollution.	4.29	0.80	0.86	
** *Affective* **				
I am concerned about the damage caused by various marine pollutants.	4.33	0.76	0.90	0.96
I should be concerned about reports of marine-related pollution.	4.26	0.80	0.87	
I should support environmental protection activities related to the marine environment.	4.34	0.78	0.90	
I agree that the government should build sanitary sewers and sewage treatment plants.	4.41	0.73	0.86	
I intend to do my part to prevent ocean pollution.	4.40	0.74	0.89	
Behavior				
(1) I will inform the relevant institutions when I notice ocean pollution.	3.80	0.87	0.81	0.951
(2) Seeing news about marine environment pollution prompts me to do more MEP research via the internet.	3.82	0.90	0.80	
(3) I do not support manufacturers or companies that pollute the sea.	4.02	0.84	0.80	
(4) I will reduce my consumption of coastal breeding seafood such as oysters.	3.77	0.99	0.62	
(5) I will reduce the use of cleaning products that pollute the sea.	4.12	0.77	0.87	
(6) I will reduce my participation in leisure activities that cause marine pollution.	4.15	0.76	0.85	
(7) I will participate in marine-related exhibitions.	4.04	0.82	0.84	
(8) I will discuss topics related to marine environment pollution with my friends.	3.94	0.89	0.83	
(9) I will participate in beach clean-up activities.	4.02	0.84	0.86	
(10) When I go to the sea, I take my refuse away with me	4.38	0.76	0.69	

**Table 3 ijerph-19-16671-t003:** Attitude-based clusters related to marine environment pollution (MEP).

Variable	Sample (*n* = 446) Mean Scores			
	Cluster 1 (52.2%; *n* = 234)	Cluster 2 (40.1%; *n* = 179)	Cluster 3 (7.4%; *n* = 33)	Average
(1) I think single-use plastics are harmful to marine ecosystems.	4.30	4.75	4.01	2.76
(2) I think marine environment pollution is currently a serious problem.	4.20	4.66	3.86	2.76
(3) I think MEP negatively affect our tourism fortunes	4.48	4.94	4.23	2.67
(4) To reduce the amount of solid waste in the ocean, we should reduce plastic production.	4.13	4.65	3.69	2.85
(5) I think daily-life waste products cause marine environment pollution.	4.29	4.84	3.87	2.64
(6) I should be concerned about the damage caused by various marine environment pollutants.	4.33	4.86	3.92	2.76
(7) I should be concerned about reports of marine-related pollution.	4.26	4.82	3.82	2.67
(8) I should support environmental protection activities related to the marine environment.	4.34	4.88	3.92	2.76
(9) I agree that the government should build sanitary sewers and sewage treatment plants.	4.41	4.91	3.99	2.73
(10) I intend to do my part to prevent the ocean pollution.	4.29	4.84	3.87	2.64

**Table 4 ijerph-19-16671-t004:** Attitudes toward marine environment pollution (MEP) and related knowledge traits.

Knowledge about MEP	Cluster 1	Cluster 2	Cluster 3	F (*p* Value)	Dunn’s Post Hoc with Bonferroni Correction
The concept of MEP	7.31	6.98	6.67	0.45 (0.641)	C1 − C2 = 0.33 (1.000) C2 − C3 = 0.31 (1.000) C1 − C3 = 0.64 (1.000)
Types of MEP	9.42	8.98	7.73	12.15 (0.00)	C1 − C2 = 0.44 (0.061) C2 − C3 = 1.25 (0.002) C1 − C3 = 1.69 (0.000)
Understanding the effects of MEP	9.01	8.91	8.11	2.58 (0.08)	C1 − C2 = 0.10 (1.000) C2 − C3 = 0.80 (0.141) C1 − C3 = 0.90 (0.071)
Understanding MEP incidents	9.10	8.66	5.15	20.78 (0.00)	C1 − C2 = 0.44 (0.000) C2 − C3 = 3.51 (0.529) C1 − C3 = 3.95 (0.000)
Expertise in MEP	4.78	4.98	4.14	1.27 (0.28)	C1 − C2 = −0.20 (1.000) C2 − C3 = 0.84 (0.350) C1 − C3 = 0.64 (0.674)
Understanding the causes of MEP	4.96	4.69	3.18	2.74 (0.07)	C1 − C2 = 0.27 (1.000) C2 − C3 = 1.51 (0.155) C1 − C3 = 1.78 (0.060)

**Table 5 ijerph-19-16671-t005:** Attitudes toward marine environment pollution (MEP) and related behavioral traits.

Behavior toward MEP	Cluster 1	Cluster 2	Cluster 3	F (*p* Value)	Dunn’s Post Hoc with Bonferroni Correction
(1) I will inform the relevant institutions when I notice ocean pollution.	4.18	3.5	2.79	71.97 (0.00)	C1 − C2 = 0.67 (0.000) C2 − C3 = 0.72 (0.000) C1 − C3 = 1.39 (0.000)
(2) Seeing news about marine pollution prompts me to do more MEP research via the internet.	4.24	3.46	2.82	80.75 (0.00)	C1 − C2 = 0.78 (0.000) C2 − C3 = 0.65 (0.000) C1 − C3 = 1.42 (0.000)
(3) I do not support manufacturers or companies that pollute the sea.	4.42	3.73	2.82	105.84 (0.00)	C1 − C2 = 0.70 (0.000) C2 − C3 = 0.91 (0.000) C1 − C3 = 1.61 (0.000)
(4) I will reduce my consumption of coastal breeding seafood such as oysters.	4.11	3.5	2.88	39.28 (0.00)	C1 − C2 = 0.61 (0.000) C2 − C3 = 0.62 (0.001) C1 − C3 = 1.23 (0.000)
(5) I will reduce the use of cleaning products that pollute the sea.	4.55	3.8	2.76	193.99 (0.00)	C1 − C2 = 0.75 (0.004) C2 − C3 = 1.04 (0.000) C1 − C3 = 1.79 (0.000)
(6) I will reduce my participation in leisure activities that cause marine pollution.	4.59	3.83	2.82	201.95 (0.00)	C1 − C2 = 0.75 (0.004) C2 − C3 = 1.01 (0.000) C1 − C3 = 1.77 (0.000)
(7) I will participate in marine-related exhibitions.	4.48	3.69	2.82	142.53 (0.00)	C1 − C2 = 0.79 (0.004) C2 − C3 = 0.88 (0.000) C1 − C3 = 1.67 (0.000)
(8) I will discuss topics related to marine pollution with my friends.	4.4	3.55	2.82	110.46 (0.00)	C1 − C2 = 0.84 (0.000) C2 − C3 = 0.74 (0.000) C1 − C3 = 1.58 (0.002)
(9) I will partake in beach clean-up activities.	4.46	3.66	2.76	139.27 (0.00)	C1 − C2 = 0.80 (0.004) C2 − C3 = 0.80 (0.000) C1 − C3 = 1.70 (0.002)
(10) When I go to the seaside, I will take my refuse away.	4.79	4.12	2.85	223.38 (0.00)	C1 − C2 = 0.68 (0.004) C2 − C3 = 1.27 (0.000) C1 − C3 = 1.95 (0.002)

**Table 6 ijerph-19-16671-t006:** Cluster profiles by demographic characteristics (%).

Demographic	Activists	Supporters	Onlookers	χ^2^ (*p* Value)
Sex				
Male	47.10%	45.20%	7.70%	14.66 (0.097)
Female	57.10%	35.70%	7.10%	
Subject				
Social Science	53.40%	35.60%	11.00%	417.72 (0.023)
Engineering	53.40%	40.70%	5.80%	
Natural science	52.30%	46.20%	1.50%	
Marine-related	68.00%	32.00%	0.00%	
Other	38.80%	44.90%	16.30%	
Year level at university				
First year	61.60%	32.60%	5.80%	16.30 (0.012)
Second year	44.20%	44.20%	11.50%	
Third year	40.90%	50.00%	9.10%	
Fourth year	45.10%	47.60%	7.30%	
Is your hometown located on the coast?				
Yes	48.00%	46.20%	5.80%	44.52 (0.104)
No	55.30%	36.40%	8.40%	
Is your university located on the coast?				
Yes	54.10%	40.60%	5.20%	413.51 (0.001)
No	45.20%	38.10%	16.70%	
Have you taken ocean-related courses related at university?				
Yes	57.90%	36.00%	6.10%	41.82 (0.396)
No	50.60%	41.60%	7.80%	
Have you participated in ocean-related activities in your university?				
Yes	56.10%	39.40%	4.50%	111.10 (0.004)
No	45.90%	41.50%	12.60%	

## Data Availability

The data that support the findings of this study are available on request from the corresponding author.

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
