# Peer review of "University Students’ Knowledge, Attitudes, and Behaviors Related to Marine Environment Pollution"

_ijerph, 2022, doi:10.3390/ijerph192416671_

Round 1
Reviewer 1 Report
The paper is an interesting attempt to use the method of a survey of university students in China to examine attitudes and behaviours related to pollution of the marine environment.
It is therefore an interesting work, carried out by interviewing the students of three universities, two of which are located on the coast and one is not. From this point of view, the research and survey work was carried out. However, according to the results presented, it seems that the authors have abandoned this initial starting point.
Also, authors must take care when writing about the spirit of the English language not to use as in a sentence line 207: ‘’We evaluated the knowledge related to MEP, we developed a ten-point scale.’’
When writing, authors must pay attention to the use of spaces between words, between words and parentheses, and finally when writing references. Authors used different reference styles, which as such were not consistent, and none of them conformed to the instructions for authors. References in the text must be written according to the instructions for authors.
The measurements need to be better described so that readers know what each value means, as in the example of Table 2, where the mean is given but it is not entirely clear what the mean is.
It is not clear from the Materials and methods that clusters are considered. However, if values for clusters are introduced into the results, then a clear basis for forming clusters should be known from the materials and methods. In addition, the clusters in Table 3 are jumbled. This needs to be clearly shown.
It is a pity that the results and the discussion do not show the attitude of the students of the two universities located on the coast in relation to the other students of the universities not located on the coast and in relation to the pollution of the marine environment.
Author Response
We greatly appreciate your careful review of our manuscript and your invaluable suggestions. We have carefully considered each of the comments and revised the manuscript accordingly.
The paper is an interesting attempt to use the method of a survey of university students in China to examine attitudes and behaviours related to pollution of the marine environment.
1.It is therefore an interesting work, carried out by interviewing the students of three universities, two of which are located on the coast and one is not. From this point of view, the research and survey work was carried out. However, according to the results presented, it seems that the authors have abandoned this initial starting point.
Response: Thank you for pointing out this problem.We have added the study limitations on page12 as follows:
“The study was only conducted in three universities in Guangdong, China. As it primarily focused on the attitude of coastal college students toward MEP, rather than with coastal students less exposed to the sea, because of the limited research sample collection, the research value is small. Future studies should consider an expanded sample that includes coastal college students and non-coastal college students to compare their MEP-related knowledge, attitudes, and behavior.”
2.Also, authors must take care when writing about the spirit of the English language not to use as in a sentence line 207: ‘’We evaluated the knowledge related to MEP, we developed a ten-point scale.’’
Response: Thank you for pointing out this problem.The sentence in line 207 has been corrected as follows:
“Knowledge related to MEP was evaluated and used to develop a ten-point scale.”.
3.When writing, authors must pay attention to the use of spaces between words, between words and parentheses, and finally when writing references. Authors used different reference styles, which as such were not consistent, and none of them conformed to the instructions for authors. References in the text must be written according to the instructions for authors.
Response: Thank you for pointing out this problem.The use of space between words, between words and parentheses, were adjusted in the manuscript.
4.The measurements need to be better described so that readers know what each value means, as in the example of Table 2, where the mean is given but it is not entirely clear what the mean is.
Response: Thank you for pointing out this problem,we have added the description of mean values on line 276-277as follows:
“We compared the average scores of participants that corresponded to each mean value in terms of cognition, emotion, and behavior toward MEP.”
5.It is not clear from the Materials and methods that clusters are considered. However, if values for clusters are introduced into the results, then a clear basis for forming clusters should be known from the materials and methods. In addition, the clusters in Table 3 are jumbled. This needs to be clearly shown.
Response: Thank you for pointing out this problem,we have added the description of forming clusters on page5:
“The two-step clustering technique was used to classify participants’ survey data based on attitudes toward MEP, using the log-likelihood distance measurement of the Schwarz Bayesian Criterion. Using this technique, the system tools divided the topics with similar load values on the MEP attitude scale into separate categories. We named these according to the overall characteristics of the topics in each category to guide the subsequent data analysis. According to the results of clustering and the comprehensive naming of similar feature topics, we obtained three cluster names of marine environmental protection attitudes: activists, supporters, and onlookers.”
6.It is a pity that the results and the discussion do not show the attitude of the students of the two universities located on the coast in relation to the other students of the universities not located on the coast and in relation to the pollution of the marine environment.
Response: Thank you for pointing out this problem.This is the limitation of this study. We have added the research limitations on page12.

Reviewer 2 Report
This is a very interesting and original article on the problem of marine environment pollution (MEP) in China, using tri-component attitude model to analyze questionnaire data from 446 randomly selected Chinese students in order to better understanding the behavior, attitude, and knowledge of university students towards MEP. Data analyses was performed with the help of the Chi-square test of independence, one-way ANOVA, and two-step 215 clustering for further data analysis. The implications of their results on the reduction of MEP were discussed and their conclusions are very relevance, namely pointing out to encouraging pro-environmental attitudes and behaviors, cultivating and develop attitudes and behaviors that lead to the conservation of the marine environment and a pro-active attitude for behavioral changes.
The article besides being a very important and timely topic, is well written and scientifically sounds good.
I, therefore, support the acceptance the article.
Author Response
We greatly appreciate your profession review of our manuscript and your invaluable suggestions.

Reviewer 3 Report
The title is accurate. The abstract is written correctly.
I have objections to the construction of the INTRODUCTION. There are four subsections. In each of them the rationale for the research is mixed together with the objectives of this study and a review of previous knowledge of the research problem. In addition, there is some individual information about the methodology, such as lines: 143-145 and the content in section 1.4 Theoretical Framework.
In my opinion, the shortcoming that is most apparent is the lack of a precisely defined main research objective and specific objectives. Therefore, I suggest a reconstruction of the INTRODUCTION structure.
First of all, please separate the subsection with the RESEARCH OBJECTIVES. Please move the methodological threads from the INTRODUCTION to the MATERIALS AND METHODS section.
Please create a section STUDY LIMITATIONS (after CONLUSIONS) and write what are the limitations.
Note to Table 1 - the percentage of students from different disciplines adds up to 100.1%.
Author Response
We greatly appreciate your careful review of our manuscript and your invaluable suggestions. We have carefully considered the comments and revised the manuscript accordingly.
The title is accurate. The abstract is written correctly.
I have objections to the construction of the INTRODUCTION. There are four subsections. In each of them the rationale for the research is mixed together with the objectives of this study and a review of previous knowledge of the research problem. In addition, there is some individual information about the methodology, such as lines: 143-145 and the content in section 1.4 Theoretical Framework.
In my opinion, the shortcoming that is most apparent is the lack of a precisely defined main research objective and specific objectives. Therefore, I suggest a reconstruction of the INTRODUCTION structure.
1.First of all, please separate the subsection with the RESEARCH OBJECTIVES. Please move the methodological threads from the INTRODUCTION to the MATERIALS AND METHODS section.
Response: Thank you for pointing out this problem.The structure of introduction was adjusted in the manuscript as you suggested.
- Please create a section STUDY LIMITATIONS (after CONLUSIONS) and write what are the limitations.
Response:
Thank you for your suggestion. We have added study limitations in “5. Research limitations ” on page 12 as follows:
“The study was only conducted in three universities in Guangdong, China. As it primarily focused on the attitude of coastal college students toward MEP, rather than with coastal students less exposed to the sea, because of the limited research sample collection, the research value is small. Future studies should consider an expanded sample that includes coastal college students and non-coastal college students to compare their MEP-related knowledge, attitudes, and behavior.”
- Note to Table 1 - the percentage of students from different disciplines adds up to 100.1%.
Response:Thank you for pointing out this problem.Regarding “the percentage of students from different disciplines adds up to 100.1%.”,the reason is the number of digits retained after rounding is different, and the displayed value is different from the actual value.

Round 2
Reviewer 3 Report
The suggested correction has been made.